# Can Use of Digital Technologies by People with Dementia Improve Self-Management and Social Participation? A Systematic Review of Effect Studies

**DOI:** 10.3390/jcm10040604

**Published:** 2021-02-05

**Authors:** David Neal, Floor van den Berg, Caroline Planting, Teake Ettema, Karin Dijkstra, Evelyn Finnema, Rose-Marie Dröes

**Affiliations:** 1Department of Psychiatry, Amsterdam University Medical Centre, Location VUMC, 1081 HJ Amsterdam, The Netherlands; t.ettema@amsterdamumc.nl; 2Department of Linguistics and English as a Second Language, University of Groningen, 9712 EK Groningen, The Netherlands; f.a.van.den.berg@rug.nl; 3Department of Research and Innovation, GGZ inGeest, 1070 BB Amsterdam, The Netherlands; c.planting@ggzingeest.nl; 4Research Group Nursing, Saxion University of Applied Sciences, 7417 DH Deventer, The Netherlands; k.dijkstra@saxion.nl; 5Health Sciences-Nursing Research, University Medical Centre Groningen, 9713 GZ Groningen, The Netherlands; e.j.finnema@umcg.nl; 6Department of Healthcare, NHL Stenden University of Applied Sciences, 8917 DD Leeuwarden, The Netherlands; 7School of Nursing, Hanze University of Applied Sciences, 9747 AS Groningen, The Netherlands

**Keywords:** dementia, mild cognitive impairment, digital technology, social health, social participation, self-management, caregiver support

## Abstract

There is increasing interest in the use of technology to support social health in dementia. The primary objective of this systematic review was to synthesize evidence of effectiveness of digital technologies used by people with dementia to improve self-management and social participation. Records published from 1 January 2007 to 9 April 2020 were identified from Pubmed, PsycInfo, Web of Science, CINAHL, and the Cochrane Central Register of Controlled Trials. Controlled interventional studies evaluating interventions based on any digital technology were included if: primary users of the technology had dementia or mild cognitive impairment (MCI); and the study reported outcomes relevant to self-management or social participation. Studies were clustered by population, intervention, and outcomes, and narrative synthesis was undertaken. Of 1394 records identified, nine met the inclusion criteria: two were deemed to be of poor methodological quality, six of fair quality, and one of good quality. Three clusters of technologies were identified: virtual reality, wearables, and software applications. We identified weak evidence that digital technologies may provide less benefit to people with dementia than people with MCI. Future research should address the methodological limitations and narrow scope of existing work. In the absence of strong evidence, clinicians and caregivers must use their judgement to appraise available technologies on a case-by-case basis.

## 1. Introduction

Social participation and the ability to manage our own lives are vital to good health, particularly to good “social health” [1]. It is, therefore, deeply concerning that social isolation is a common experience for people living with dementia [2,3]. There are many factors that contribute to social isolation, including stigma, loss of independence, and inaccessibility of public spaces [4,5,6]. In addition to addressing factors that cause or exacerbate social isolation, solutions to facilitate good social health in dementia have been made a global priority [7]. A particular area of interest in addressing social health in dementia has been the use of digital technologies to facilitate self-management and social participation [7,8,9,10].

The importance of social health and the consequences for people with dementia of insufficient social participation have been highlighted by the impacts of “lockdown” measures in response to the 2020 COVID-19 pandemic [11]. The pandemic has also increased the urgency of implementing effective technological solutions to overcome the restrictions on physical interaction and the pressure on health and care systems. Governments in some countries, including the Netherlands, have made emergency funding available to health and care institutions to invest in digital technologies in response to these challenges [12].

Despite intensifying interest and investment, previous reviews of the literature have failed to consistently demonstrate high-quality evidence of the effectiveness of technological solutions in improving cognitive functioning, self-management, or social participation for people with dementia or mild cognitive impairment (MCI) [13,14,15]. At the same time, advances in digital technology continue at an exponential rate and the application of technologies aiming to improve social participation and self-management is a rapidly growing commercial and scientific field [16,17]. To gain insight into the factors that actually work well and those that do not, this review aims to capture the latest scientific evidence of the effectiveness of digital technologies for facilitating self-management and social participation for people with dementia.

The concept of social health has been defined in the context of dementia as the capacity to fulfil one’s potential and obligations, management of one’s own life, and participation in meaningful and social activities [18]. This review was conducted within the scope of a European project to evaluate technology for self-management and social participation [19], and therefore focuses on these latter two components of social health. By a European consensus, self-management has been defined as “the ability to preserve autonomy and to solve problems in daily life, as well as to adapt to and cope with the practical and emotional consequences of dementia” [18]. Social participation has been defined as “the act of being occupied or involved with meaningful activities and social interactions and having social ties and relationships, which are meaningful to the person living with dementia themselves”. 

A distinction can be made between technologies based on the intended user of the technology—either health and care professionals or people with dementia themselves [20]. In this review, we focus on technologies that are used by people with dementia themselves. This represents the most direct form of support for self-management. Such technologies may include software applications or “apps” designed to be used by the person with dementia, or include specific hardware components such as tablet computers, virtual reality (VR) devices, or wearable technologies. This review does not consider so-called “ambient” technologies that require no interaction of the person with dementia (such as fall monitors installed in homes or other “smart home” technologies) or technologies designed to be used only by caregivers or health and care professionals (such as medical devices). A distinction can also be made between people experiencing impaired cognition with a diagnosis of dementia and people with MCI. In this study, we are interested in both dementia and MCI. The primary objective of this study is to identify and evaluate effect studies of interventions based on the use of digital technologies by people with dementia or MCI to facilitate self-management and social participation.

The use of digital technologies by people with dementia or MCI to improve social health is complex. Many people living with dementia receive some form of help, care, or support with certain activities. This may be from a professional or an informal caregiver, such as a spouse, family member, or volunteer. In our experience conducting research on technology used by people with dementia or MCI, informal caregivers often play an important role in the effective use of technology, however the nature and quality of interactions can vary. The relationship between the person with dementia and caregiver may, therefore, be an important contextual factor, alongside the design, content, and implementation of the technology, which influence the effectiveness of the intervention [18]. The caregiver characteristics and outcomes should, therefore, ideally be taken into account when evaluating such an intervention [21,22]. In acknowledgement of these complexities, there are two further secondary objectives of this review: firstly, to identify the relationship between the caregiver and the person with dementia or MCI, as well as outcomes relating to caregivers in included studies; secondly, to identify any facilitators and barriers to the implementation of technology in the included studies.

## 2. Materials and Methods

This systematic review was carried out with reference to the PRISMA guidelines and best practices as outlined by the Centre for Reviews and Dissemination [23].

### 2.1. Data Sources and Search Strategy

A systematic search strategy was developed by D.N. and C.P. Searches were conducted of the following databases, within the date range of 1 January 2007 to 9 April 2020: Pubmed, APA PsycInfo, Web of Science, Cumulative Index of Nursing and Allied Health Literature (CINAHL), and the Cochrane Central Register of Controlled Trials. To identify non-indexed and indexed articles, MeSH terms, thesaurus terms, and non-MeSH terms were included. The full search strings used are available in Appendix A. Search terms related to “dementia”, “computer-assisted technology”, and “social health”. Results of the searches were exported to appropriate reference management software and de-duplicated before undergoing screening. Where trial registry entries were identified but there were no published results listed in the registry, a further search using Google Scholar was conducted using the trial registration number as the search term, and any articles identified were screened for inclusion in the review. The protocol for this review was not published in advance.

### 2.2. Inclusion and Exclusion Criteria

Inclusion criteria were controlled effect studies into digital technology interventions focused primarily at people with a diagnosis of dementia or MCI and published in 2007 or later.

The year 2007 was chosen because this is the year that the first iPhone smart phone was launched. We limited the study to controlled interventional studies, as such studies can provide strong evidence for drawing causal inferences about the effectiveness of interventions compared to non-controlled or observational studies. We did not specify what the nature of the control intervention should be. Primary target-users of the technological intervention in included studies were adults with a diagnosis of dementia (of any etiology) or with MCI. MCI is diagnosed on the basis of objective cognitive impairment, without significant impact on daily life and activities [24]. However, between two and four of every ten people with MCI go on to develop dementia, and we expected that many studies would have included participants with MCI in addition to people with dementia. An intervention met the definition of a digital technology if it was inherently dependent on any electronic device that comprised, or interfaced with, any kind of computer. Self-management and social participation are both aspects of daily life, and therefore we decided to include only studies that assessed outcomes in an ecologically valid setting in this review. We excluded studies that reported on outcomes theoretically related to social participation or self-management, but only using measurements obtained under laboratory conditions.

To be included in this review, the authors of the original papers did not have to explicitly name “self-management” or “social participation” as outcomes of objectives of the study. The reviewers included all studies that reported outcome measures relevant to self-management or social participation based on the operational definitions noted in the introduction to this article. With respect to measures that might be relevant to self-management, we decided not to include studies that only reported on basic activities of daily living (BADL). Whilst BADL are relevant to self-management, a measure of BADL alone is not sufficient to meet the operationalized definition of self-management in dementia. Studies that reported on instrumental activities of daily living (IADL) or measures of autonomy or self-efficacy in addition to, or instead of, BADL were considered for inclusion in this review. Studies in which the person with dementia was supported in the use of the digital technology by a formal or informal caregiver were also included, as were studies in which the technology was used by the caregiver independently of the person with dementia, as long as the person with dementia also used the digital technology.

Accordingly, records were excluded from this review if they were: (1) published before 2007; (2) not reporting on an interventional study; (3) not reporting on a controlled study; (4) not reporting on participants with dementia or MCI; (5) not reporting on an intervention that met the definition of a digital technology; (6) reporting on an intervention that was exclusively a medical or diagnostic device; (7) reporting on a technology that the people with dementia or MCI did not use themselves; (8) reported on an intervention that was not aimed at improving self-management or social participation; (9) reported on outcomes that were only measured in a laboratory environment; or (10) only reporting on BADL in relation to self-management (not complemented by any measure of IADL, autonomy, independence, self-efficacy, or some other aspect of self-management).

### 2.3. Study Selection and Data Extraction

All references identified were screened independently by one junior and one senior researcher (D.N. or F.v.d.B.; and E.F., R.M.D., K.D., or T.E.). A first round of screening took place on the basis of the published title and abstract. Studies not excluded on the basis of abstract screening were screened on the basis of full texts. If consensus between two independent reviewers could not be reached, a third independent reviewer was available to make a definitive decision as to whether the reference should be included in the review. Where required, translation from languages other than English and Dutch was carried out by qualified translators.

Data were extracted from included articles by means of a standardized data extraction form. This form was piloted and reviewed after the extraction of data from the first study, resulting in minor revisions. Data extraction from all included studies was performed by a junior researcher (D.N.) and checked by a senior researcher (R.M.D. or T.E.). General study characteristics extracted included the study design, sample size, setting, and population characteristics (age, gender, and health status or diagnosis). With respect to the primary focus of the review, extracted data described: the nature of the digital technology used (relevant hardware and software components); the intended function of the technology within the context of the intervention; the intended user of the technology; the outcomes measured, which related to social participation or self-management (including specific instruments used); and the results of the intervention. In line with the secondary focus of the review, data were also extracted concerning: any statements relating to facilitators or barriers experienced in implementing the intervention in the context of the study; and whether any caregiver outcomes were measured (and if so, the results).

### 2.4. Methodological Quality and Weight of Evidence

Weight of evidence was assessed on the basis of the Evidence for Policy and Practice Information (EPPI) weight-of-evidence approach (WoE) [25]. This approach provides a framework for weight-of-evidence assessment, which breaks down the assessment into four judgements. The intrinsic quality of included studies (WoE A) is a judgement of the quality of methodological design and risk of bias in the original study. The appropriateness of the method in the context of the review (WoE B) is a judgement of the extent to which the research design of the original study is appropriate to address the current review question. The relevance of each study to the review question (WoE C) is a judgement of how closely the original study population, intervention, and outcome measures match those of the current review question. Finally, an overall judgement is made, taking into account the three judgements named above (WoE D). The WoE framework can be combined with tools to conduct the individual WoE assessments. In this review, we used the National Institute of Health (NIH) tool for the assessment of randomized controlled studies to provide structure to our assessment of the internal validity of the included studies (WoE A) [26]. The weight-of-evidence assessment for all included studies was conducted independently by two junior reviewers (D.N. and F.v.d.B.). Judgements WoE A could be judged as good, fair, or poor. WoE B to WoE D were judged as high, fair, or low. These assessments were checked with a senior researcher (RMD) and any disagreements were resolved.

### 2.5. Synthesis

We anticipated considerable heterogeneity of interventions, study protocols, and outcome measures between studies included in this review, as the scope of the cited definitions of self-management and social participation was broad. Similarly, many different digital technologies exist, which could in theory be applied to the facilitation of social participation and self-management, from personal computers and touch screen devices, to wearable technologies, to software applications. The data synthesis was, therefore, undertaken by means of a narrative synthesis approach, which is well-suited to the synthesis of heterogeneous literature [27]. Included studies were grouped into clusters on the basis of the extracted data regarding population, nature of the technology, and intervention components and outcomes measured. With respect to studies analyzing the same interventions or the same outcomes, descriptions of the presence and direction of effects were noted. We did not plan to conduct any further statistical meta-analysis because of the expected heterogeneity in technological interventions and small number of included studies.

To categorize any statements made on facilitators and barriers of implementation of the technologies, we used the UK Medical Research Council (MRC) guidance for conducting process evaluations. This guidance defines several factors that can influence the final outcomes of intervention studies, including the context (the social and physical environment in which the intervention is received), implementation of the intervention (what participants actually receive), and mechanisms of impact (participants’ responses and interactions with the intervention, mediators, and unexpected pathways leading to the outcomes measured) [22].

## 3. Results

The literature search across the databases returned 2135 results. Figure 1 illustrates the screening process. After deduplication, 1394 unique records remained. Initial title screening was carried out by D.N. to remove any records that were irrelevant to this review. This resulted in the removal of 424 records. The remaining 970 records were screened on the basis of their associated abstract—and where applicable, associated trial registry information—by two reviewers independently. Forty-six records were identified based on the abstract as appropriate for assessment by a close reading of the full text. One further record was identified for full-text screening as a separate abstract was not available.

Of the 47 full texts that were read, only nine studies met all of our inclusion criteria [28,29,30,31,32,33,34,35,36]. Sixteen records were excluded because they were determined not to be reporting results from controlled interventional studies [37,38,39,40,41,42,43,44,45,46,47,48,49,50,51,52]. Eight records were excluded because outcomes of the intervention were only measured in a laboratory environment [53,54,55,56,57,58,59,60]. Six records were excluded because they did not report any outcomes that were directly related to social participation or self-management [61,62,63,64,65,66]. Four records reported on BADLs only [67,68,69,70]. Three studies were excluded because although participants with some form of cognitive impairment were included in the study, the outcomes were not reported separately from outcomes from those who were either cognitively healthy or had other pathologies [71,72,73]. One record was excluded because it was discovered to be a protocol only, not reporting any results [74]. 

### 3.1. Characteristics of Included Studies

The aim and key elements of the included interventions are described below. Details regarding the design and methods of the included studies are shown in Table 1. As anticipated, the studies that we identified were heterogenous with respect to the specific target population, nature of the intervention, control group intervention, and outcomes measured. Most challenging to interpret were the results of the three studies in which there was also heterogeneity within the studies themselves, because participants with formal diagnoses of dementia or MCI were included and analyzed together with participants with subjective cognitive impairment, objective but non-diagnostic cognitive impairment, or even non-cognitively impaired participants [29,31,36].

Each study included in this review evaluated a different intervention. We identified three technology-related clusters to which the experimental interventions belonged.
Virtual reality (VR)-based interventions
-Combined physical and cognitive VR-based training [31]: The primary aims of the intervention were to improve the user’s cognitive function and reduce neuronal oxidative stress. A VR-based program combining aerobic exercise and cognitive training was administered over 6 weeks, in three sessions per week (18 sessions total). Each session was of 40–45 min duration, in which participants interacted with three virtual environments (15–20 min cycle in a virtual park, 5 min crossing at virtual crossroads, 20 min shopping in a virtual supermarket).-VR-based physical and cognitive training [29]: The primary aims of the intervention were to improve the user’s cognitive function and performance of IADL. The Microsoft Kinect system was used to engage participants in Tai Chi, resistance and aerobic exercises, and simulated functional tasks, such as window cleaning and stair climbing. The HTC VIVE system (VR glasses) was used to engage participants in VR games based on IADL (such as shopping and food preparation). The intervention was delivered in sessions (three sessions per week for 12 weeks, 36 sessions in total). Each session lasted one hour (40 min physical activity, 20 min cognitive training).-VR-based non-specific computer training [33]: The aims of the intervention were to improve users’ cognitive function and quality of life. Users were trained to use the Nintendo Wii to engage in virtual reality sports activities, including table tennis, fencing, and archery. The intervention was delivered by occupational therapists, during three sessions per week for ten weeks, of 10 min duration.Other wearable technologies
-Social Support Aid, a web-based mobile app with a smartwatch [30]: The primary aim of the intervention was to improve the users’ social engagement. Social Support Aid is a mobile phone-based app, which connects to a smart watch. Facial recognition software is used to assist people with dementia in the identification (names and relationships) of people that they interact with. Participants were given the technology to use for 6 months and were free to use it as much or as little as they chose.-SenseCam, a wearable camera [35]: The primary aims of the intervention were to improve the cognitive function (specifically episodic and autobiographical memory), and the wellbeing and quality of life of the user. The SenseCam wearable camera was worn by participants around the neck while performing routine activities. The images captured were reviewed and discussed with a psychologist at interval appointments (face-to-face, twice per week). Participants were given SenseCam to wear for six weeks and were encouraged to wear the camera every day for the longest time possible.Software applications
-Tablet-based cognitive training and rehabilitation [34]: The primary aims of the intervention were to improve cognitive function and functional capabilities of the user, and to unburden the user’s informal caregiver. A tablet computer was used for guided and independent cognitive training. Participants also engaged in a wellbeing program via the tablet, with videocalls for peer support. The telerehabilitation program was administered over 13 weeks—guided for the first 9 weeks, then completed independently for the remaining 4 weeks. Participants engaged in seven face-to-face sessions (four sessions of 4–6 h each for patients and family members together, and three peer support sessions of 2 h each separately for patients and family members).-Electric calendar [32]: The primary aims of the intervention were to improve the user’s cognitive function and reduce behavioral disturbances by means of a software application for an Android tablet with page-a-day calendar, clock, and alarm function for scheduled “events”. The intention was that the person with dementia or MCI would regularly view the calendar and interact with alarms. The calendar was set-up and kept up to date by a caregiver. The calendar was continuously present in the participant’s home for 12 weeks.-WESIHAT 2.0©, a web-based health education tool [36]: The primary aim of the intervention was to improve cognitive function of the user. Secondary aims were to improve users’ functional capabilities, social support, mood, and quality of life, and to reduce loneliness. A web-based application with four components: (1) a screening tool for risk of memory impairment; (2) lifestyle advice for promoting memory and health; (3) health diary; (4) healthy food menu, with meal preparation tips, shopping guidelines, and a nutrition-related quiz. Participants were exposed to the application in sessions: four sessions per week for six months (at least 100 sessions total). Sessions lasted at least 30 min.-Computerized errorless learning program (CELP) for memory training [28]: The primary aim of the intervention was to improve the cognitive function of the user. Improving mood and functional capabilities were secondary aims. The memory training included several components: basic training on various memory types; the use of mnemonics and learning principles, as well as name–face association; and advanced memory training and strategies for applying memory techniques to ADLs. Participants received the intervention in sessions: approximately two sessions per week up to a total of twelve sessions. Each session lasted around 30 min.

### 3.2. Quality and Weight-of-Evidence Assessment of Included Studies

Table 2 summarizes the quality assessment and weight of evidence of the included studies.

Only one of the included studies was found to be of good methodological quality [33]. Six studies, considered to be of a good or fair methodological quality, reported outcomes relevant to self-management [29,31,33,34,35,36]. Four studies of a good or fair methodological quality reported outcomes relevant to social participation [30,33,35,36]. In our assessment, all of the studies faced methodological issues relating to small sample sizes (range *n* = 10 to *n* = 78) and effectively blinding participants, investigators, or both to group assignment. Only three studies clearly reported that an intention to treat analysis was conducted [30,33,34]. 

None of the studies that we identified were judged to be of high relevance to the primary review question. Only three of the studies investigated outcomes that were relevant to both self-management and social participation [33,35,36]. None of the studies described their primary outcomes as “self-management” or “social participation”, or cited the same definitions of those terms that we used to design this review. In most of the studies identified, outcomes relevant to self-management or social participation were considered secondary, most commonly to a primary outcome of cognitive function. 

Seven of the nine included studies were judged to be of a fair overall weight of evidence. These studies were included in the narrative synthesis [29,30,31,33,34,35,36]. The two studies judged to be of poor quality and of overall low weight of evidence pertained to software applications, and evaluated outcomes relevant to self-management but not social participation [28,32]. Neither reported quantitative, statistically significant effects of the interventions evaluated compared to the control group. Because of their low quality and low weight of evidence, they were excluded from further synthesis with respect to the self-management outcomes reported.

### 3.3. Results of Included Studies

The reported results with respect to self-management, social participation, and caregiver outcomes are shown in Table 3. 

Of the studies reporting self-management outcomes, all reported on IADL but none reported on other aspects of self-management, such as self-efficacy or experienced autonomy [28,29,31,32,33,34,35,36]. Three studies reported no statistically significant effects of the intervention compared to the control group [28,31,34], four reported positive effects of the intervention compared to the control group [29,33,35,36], and none reported significant negative effects of intervention compared to control. One study reported only qualitative outcomes [32]. Effect sizes of positive results with respect to self-management outcomes were large, according to Cohen’s benchmarks (ƞ^2^p = 0.191, ƞ^2^p = 0.217, ƞ^2^p = 0.29, and ƞ^2^ = 0.821, as measured by WHODAS, LIADL, IAFAI, and SF-36, respectively) [75]. To identify possible trends and generate hypotheses for future research, we clustered the studies of good or fair weight of evidence, which evaluated self-management outcomes, with respect to the target-users of the intervention and with respect to the technology used. With respect to the target population, four distinct “clusters” were identified: users diagnosed with dementia (*n* = 2 studies, both finding no significant effects of interventions [34,35]); users diagnosed with MCI or dementia (*n* = 1 study, finding no significant effects of intervention [31]); users diagnosed with MCI (*n* = 2 studies, both finding a positive effect of the intervention [29,33]); and users diagnosed with dementia or MCI, or with no formally diagnosed cognitive impairment (*n* = 1 study, finding a positive effect of the intervention [36]). The three clusters with respect to the nature of the digital technology on which the interventions were based were: VR (*n* = 3 studies, one reporting no significant effects of the intervention [31], two reporting a positive effect of the intervention [29,33]); wearables (*n* = 1 study, reporting a positive effect of the intervention [35]); and software applications (*n* = 2 studies, one reporting no significant effects of the intervention [34], one reporting positive effects of the intervention [36]).

As shown in Table 1 and Table 2, four of the included studies evaluated outcomes with respect to the social participation of target-users [30,33,35,36]. In three cases [30,33,35], no significant effects of the intervention were reported and one study identified positive effects of the intervention [36]. None of the included studies reported significant negative effects of the intervention. The effect sizes of the positive results (ƞ^2^p = 0.123 and ƞ^2^p = 0.191 with respect to outcomes measured by the MOSS and WHODAS instruments, respectively) were reasonably large [75]. Because the number of clusters with respect to target-users and types of technology would have been larger than the number of included studies, clustering was not undertaken with respect to social participation outcomes.

Only one of the included studies reported outcomes for informal caregivers of the target-user (positive effect of the intervention on one subscale of the COPE instrument, effect size not reported) [34].

### 3.4. Facilitators and Barriers to Implementation

Table 4 comprises the extracted statements regarding facilitators and barriers to implementation.

None of the included articles made reference to the MRC Guidance for process evaluation and the statements we extracted were not pre-categorized by the original authors. For the purposes of this synthesis, we clustered statements under the three guidance categories [22].

Most statements made by the authors of the articles in this review, about both facilitators and barriers to implementation, were categorized as concerning the mechanisms of impact of the intervention. Six of the eight statements regarding facilitators mentioned positive emotional responses of participants to the intervention being evaluated. Two statements regarding barriers to implementation cited the complexity of the technology for the user as an important barrier, while two cited perceived functional limitations of the technology evaluated. Of the four statements categorized as relating to implementation, two concerned the nature and frequency of support sessions that were provided to participants. Although all of the studies reported on background characteristics of included participants, only one statement explicitly acknowledged the participants’ backgrounds as an important contextual factor that may have influenced implementation of the intervention.

## 4. Discussion

This systematic review identified controlled studies evaluating the effectiveness of digital technologies used by people with dementia to improve their self-management or social participation. However, the scope of the available evidence to support answering the review question is limited. We note three important features of the records identified: significant heterogeneity, limited relevance to the review question, and limitations in methodological quality.

We only identified a small number of studies, within which the technologies evaluated could be categorized into three clusters: VR-based technologies, other wearable technologies, and software applications. However, the characteristics of the participants, interventions, and outcome measures differed between studies, such that none of the studies were directly comparable to any other study. We also noted that the nature of the control interventions varied between the included studies. This might be expected to affect the reported results with respect to between-group differences. Of the seven studies that we assessed as being of good or fair weight of evidence, three studies included an active, non-technological control group, one study included an active technological control group, and three had an inactive control intervention (defined as continuing usual care, no treatment, or waiting list). Counterintuitively, all four of the studies with an active control demonstrated at least one statistically significant, positive effect of the intervention with respect to self-management or social participation, whereas the three studies that were compared to no active control failed to demonstrate any statistically significant differences. One explanation for this could be that the statistical power of the studies with no active control groups was lower. The sample sizes of six of the studies were similar (*n* = 42 to *n* = 78) but one of the studies which had no active control was very small (*n* = 10). Alternatively, given that the results concern only six studies, this unexpected result could simply be down to chance.

Only one of the studies which reported positive results was of good methodological quality. For the other studies, the most notable issues were with sample size and power, inadequate blinding of participants or investigators, and inadequate statistical analysis. It is particularly noteworthy that so few of the investigators reported intention-to-treat analyses. Whilst intention-to-treat is well-established as the standard method of analyses for RCTs, previous studies have found that the majority of published RCT results lack intention-to-treat analysis [76]. All studies were judged to be of low relevance to this review question, because none of the studies used the definitions of self-management or social participation that we used for this review and because outcomes measured that were relevant to this review were mostly considered secondary (most commonly to cognitive function). It is an important distinction that most of the relevant outcomes were considered secondary, because study protocols and statistical power are generally optimized with respect to the primary rather than secondary outcomes of a study. Given that the definitions of self-management and social participation that we used were only published in 2017, it is not surprising that few investigators cited these definitions. However, this does not explain why so few studies were identified in which outcomes relevant to self-management and social participation were considered primary. It may be that investigators did not consider self-management or social participation a priority, however an alternative explanation would be that the investigators considered these outcomes dependent on a more fundamental underlying mechanism, such as improved cognition.

Overall, we identified studies of fair weight of evidence, reporting positive outcomes of the respective interventions evaluated, with large effect sizes reported. A comparable number of studies that were also of fair weight of evidence found no effect of the evaluated interventions compared to control groups. That none of the included studies reported negative effects of the interventions with respect to either self-management or social participation should not be over-interpreted. It may be that digital technologies have at least neutral effects on self-management outcomes, but other explanations, such as publication bias, could easily account for this finding. A non-publication rate of around 27% of RCT results in the field of digital health has been reported, which whilst better than in some fields of biomedical research, remains significant [77]. It is also noteworthy that two of the three studies that reported intention-to-treat analyses failed to demonstrate any between-groups differences on post-test outcomes. Overall, given the heterogeneity between included studies, their mostly fair methodological quality, and their low relevance to this review, we conclude there is weak evidence to suggest that technologies used by people with dementia and MCI may be able to help improve self-management. There is insufficient evidence at this time to draw conclusions about the impact of technology use on social participation of people with dementia or MCI.

To help researchers prioritize participant groups or interventions in the future, we attempted to cluster studies on the basis of the cognitive status of participants, the control condition, and the nature of the intervention. Owing to the small number of studies included, we were largely unable to identify trends with respect to the outcomes per cluster. However, with respect to self-management outcomes, we note that the three studies that found positive effects of the technology included, respectively, only users with MCI and users with MCI and less severe cognitive complaints (in addition to users with dementia), whereas the two interventions that targeted only target-users with formally diagnosed dementia reported no effects of the interventions. From the limited evidence that we have identified, we would tentatively hypothesize that people with a confirmed diagnosis of dementia might benefit less from using the technologies identified in this review compared to target-users with mild cognitive impairment.

A secondary objective of this review was to identify and evaluate outcomes for informal caregivers of technology used by people with dementia. We were unable to identify sufficient evidence from included studies to draw conclusions regarding outcomes of the interventions for informal caregivers. The lack of evidence might be due to investigators choosing to investigate or report caregiver outcomes separately, although we did not note any references to caregiver outcomes reported elsewhere. An alternative explanation could be that investigators deliberately chose not to investigate caregiver outcomes in parallel with outcomes for the person with dementia, perhaps because it was considered too demanding—either on their own resources or on participants.

We also set out to characterize facilitators and barriers to implementing interventions identified in previous studies. We consider it a further methodological limitation of most of the identified studies that a well-documented process evaluation, carried out alongside the RCT, was lacking. It is very important to understand which factors may have influenced the study outcomes [21,22]. Where statements about facilitators and barriers were extracted, the extent to which participants enjoyed using the technology seemed to be a factor that investigators considered important. This is presumably based on the assumption that enjoyment in using the technology would facilitate regular use, without which the intervention could not be expected to affect the outcomes measured. This may be intuitive but there are many more factors than enjoyment alone, which may have a significant modifying effect on the impact of the trial [22]. Statements made regarding complexity or limited functionality of the technology raise the question as to whether the preceding design process was sufficiently user-centered. It might be hoped that such limitations would be discovered and avoided during the design of an intervention, before formal evaluation in an RCT.

The statements regarding implementation factors provide useful information for researchers who wish to undertake further research with the interventions in question, namely how the protocol for implementing the intervention might be modified or optimized. It is, therefore, surprising that so few statements were identified. It might be that the authors of other studies did not consider the manner in which the intervention was implemented to be of importance. Alternatively, it is possible that other authors did not consider that there were any feasible modifications that could be made to the implementation of the intervention.

The lack of statements explicitly addressing contextual facilitators and barriers could be because most authors felt that there was sufficient contextual information provided in their article for readers to draw their own conclusions about the possible impact of the context on the results of the reported study. It is highly unlikely that no contextual facilitators or barriers were encountered during the other studies, however in some cases it is possible that the authors did not consider the context to have had a sufficiently large impact on their evaluation of the intervention to be worthy of reporting.

In summary, our review shows that the availability of high-quality evidence in this field does not seem to have significantly progressed from previous reviews, despite rapid growth in interest in, and spending on, technological solutions for supporting social health in dementia [7,8,9,10,12,13,14,15].

### 4.1. Strengths and Limitations of This Review

The most important strength of this review was the use of a systematic search strategy. We searched for, screened, and included studies published in any language. We also made use of both MeSH headings and multiple synonymous search terms in order to identify as many relevant articles as possible from the databases searched. At each stage of the screening process, all records were independently screened by at least two reviewers to avoid bias in the results.

A limitation of this review is that the execution of the search strategy was limited to English language databases, although we did not exclude any of the articles we identified just because they were not published in English. We did not include grey literature in this review, nor did we undertake a specific effort to identify possible unpublished results. We cannot, therefore, exclude the possibility that some relevant evidence was absent from our results. A further limitation stems from our decision to base this review on the definitions of social health, self-management, and social participation, as previously cited. Whilst these definitions are present in the literature, there may be alternative definitions of these concepts of which we are not aware or alternative terminology used by other investigators with the same definitions, meaning our list of search terms may, therefore, not have been sufficiently exhaustive to identify all relevant records. Finally, in the absence of published literature or recommendations, we determined as a group which instruments were relevant to the cited definitions of self-management and social participation. Other investigators may have made different judgements regarding the relevance of certain instruments and could, therefore, have made different decisions about which of the identified records should have been included in this review.

### 4.2. Recommendations for Future Research

Given the systematic methodological limitations of the studies identified, a fundamental question is if the RCT is the most appropriate methodological approach to evaluating these kinds of interventions. It has previously been pointed out in the literature that traditional methods of evaluation might be too time-consuming to usefully evaluate digital mental health applications [78]. It may be that there are feasible alternative forms of evaluation to the traditional RCT that would be acceptable to policy-makers, payers, and providers of health and social care. The level of evidence considered necessary and sufficient by different stakeholders for different purposes should be further investigated.

Whenever RCTs are conducted in this field, more needs to be done to ensure that studies are of a good methodological quality. Funding organizations and investigators must ensure that adequate resources are available to conduct adequately powered studies. Investigators should also improve the quality of RCTs in this field by publishing protocols in advance, conducting intention-to-treat analyses, and considering novel ways of blinding participants. In our own experience, blinding participants and investigators to group assignment is currently always a challenge when assessing technological interventions. Only one of the studies that we identified that used an “active” control group also used a technological control intervention. In clinical drug trials placebos can be used, which exactly resemble the active intervention but have no theoretical mechanism by which to improve outcomes. One might speculate that as this field develops, it may be helpful and feasible to use either a digital placebo or a “digital care as usual” condition, where care as usual includes digitally delivered components. A digital control condition would more closely resemble the experimental intervention than a non-technological control intervention. It should, therefore, be easier for investigators to conceal group assignments from participants and investigators alike. It has previously been speculated that a “digital placebo effect” exists [79]. Designs including a digital control arm should have the additional benefit of controlling more effectively for such an effect than studies that do not include a digital control arm. This would be one way of systematically improving the quality of future RCTs in this field.

We recommend that future research should focus on the most important gaps that we have identified in the current literature. We would urge investigators to make self-management and social participation, as defined in the literature, primary outcomes of their studies. Good social health is a vital component of good general health [1,7]. Clinicians, payers, and policy-makers need evidence to make informed decisions in the face of the growing availability of technological interventions [11,12,16,17]. To comprehensively assess self-management, we advise that investigators should use appropriate validated instruments to assess self-efficacy and experienced autonomy in addition to function or IADL [18]. We would also advise investigators to consider evaluating outcomes for informal caregivers, even where they are not the primary intended user of the technology, since the close relationships between people with dementia and informal caregivers are likely to moderate effects of interventions, and vice versa. Finally, we would urge investigators to conduct and report formal process evaluations and to identify facilitators and barriers that might influence study results. We recommend that investigators refer to published guidance, such as the UK Medical Research Council (MRC) guidance for conducting process evaluations [20,21].

## 5. Conclusions

The limited quantity and quality of the evidence does not allow for strong conclusions or recommendations for clinical practice with respect to the use of digital technologies by people with dementia or MCI for improving self-management or social participation. Rather, we have found that there is great scope to improve the scientific rigor and to expand the evidence base with respect to technologies aiming to enhance the self-management and social participation of people with dementia. We hope and expect that given the pressures created by the rapidly increasing number of people living with dementia, the global COVID-19 pandemic, and the limited ability of traditional health and social care systems to meet people’s needs, there will be rapid growth in this field in the years to come. In the meantime, healthcare professionals and informal caregivers must rely on their best judgement to evaluate the potential benefits and limitations of digital technologies for people with dementia or MCI on a case-by-case basis.

## Figures and Tables

**Figure 1 jcm-10-00604-f001:**
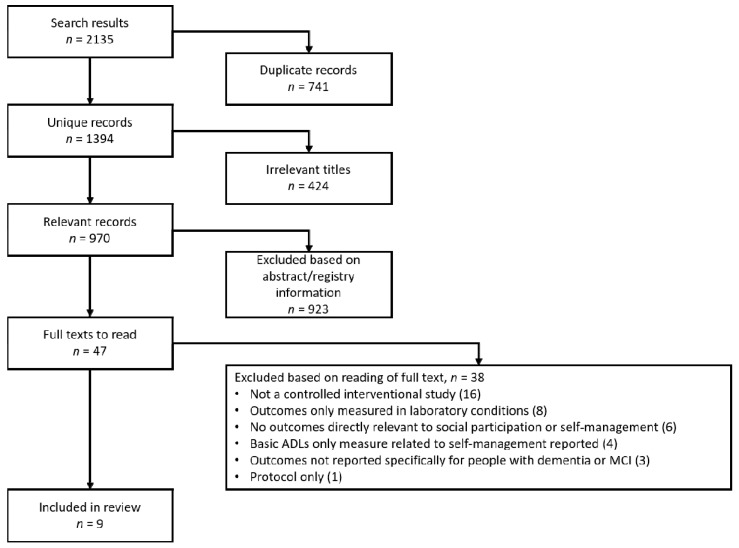
Search result screening process. ADLs: Activities of Daily Living; MCI: Mild Cognitive Impairment.

**Table 1 jcm-10-00604-t001:** Design and methods of included studies.

Authors, Year, and Language of Publication	Setting and Reported Funding Source	Study Design and Sample Size	Experimental Intervention	Control or Comparison Intervention	Sample Characteristics	Outcome Measures Relevant to Self-Management	Outcome Measures Relevant to Social Participation	Outcome Measures Related to Caregivers
Lee et al. 2013 (English). [28]	Hong Kong. Both outpatient and inpatient; urban setting. Source of funding not reported.	Single-center, single-blinded, three-arm, randomized controlled trial (RCT). Measurements at baseline, 6 weeks, and 3 months; *n* = 24 (7 experimental; 6 offline training; 6 control).	Computerized errorless memory training program (CELP).	(Therapist-led errorless learning programme) TELP arm: Therapist-led memory training Control arm: waiting list.	Persons with ‘early dementia’ (Clinical Dementia Rating Scale [CDR] = 1); Mean age: 77.7 ± 6.07; 6 male, 13 female. Mean Mini Mental-State Examination (MMSE) CELP group 15.3 ± 2.7; mean MMSE TELP group 17.0 ± 3.5; mean MMSE control 17.6 ± 4.7	Hong Kong Lawton Instrumental Activities of Daily Living Scale (HKLIADL)	None	None
Liao et al. 2020 (English). [29]	Taiwan. Community-dwelling, recruited through community and day care centers. Funding from Taiwan Ministry of Science and Technology.	Single-center, single-blinded, two-arm RCT. Measurements at baseline and after 12 weeks; *n* = 42 (21 experimental; 21 control).	Microsoft Kinect VR system, for Tai Chi, resistance, and aerobic exercises.	Combined physical and cognitive training	Older adults with MCI (MMSE ≥ 24, Montreal Cognitive Assessment [MoCA] < 26); VR group: mean age 75.5 ± 5.2, 11 female, 7 male; control group: mean age 73.1 ± 6.8; 12 female, 4 male.	Lawton Instrumental Activities of Daily Living scale (LIADL)	None	None
McCarron et al.2019 (English). [30]	United States of America. Described as community setting. Funding from United States National Institute on Aging to Advanced Medical Electronics.	Pilot single-center, non-blinded, two-arm RCT. Measurements at baseline, 3 months, and 6 months; *n* = 48 (20 experimental; 28 control).	Social Support Aid (SSA) mobile phone-based facial recognition application.	Usual care	Persons diagnosed with dementia (*n* = 29), persons with self-reported memory loss or concerns (*n* = 19). Mean age 74.90 ± 6.98; 23 male, 25 female.	None	Self-reported satisfaction with social contacts. Pleasant Events Schedule—Alzheimer’s Disease (PES-AD); Dementia Quality of Life (DQoL).	None
Mrakic-Sposta et al. 2018 (English). [31]	Italy. Not further described. Funding from the Italian Ministry of Education, Universities, and Research (MIUR).	Pilot single-center, non-blinded, two-arm RCT. Measurements at baseline and after 6 weeks; *n* = 10 (5 experimental; 5 control).	VR-based program combining aerobic exercise and cognitive training.	No treatment	Persons described as having MCI (*n* = 6), described as mild dementia (*n* = 4). Mean age: 73.3 ± 5.7 years; MMSE 23 ± 3.4; 4 male, 6 female.	Functional Activities Questionnaire (FAQ)	None	None
Nishihura et al. 2019 (English). [32]	Japan. Community dwelling participants recruited via day care center. Funding from the Japan Society for the Promotion of Science.	Single-centre, non-blinded, randomized cross-over study. Measurements at baseline, after 12 weeks, and after 24 weeks; *n* = 27 (15 experimental; 12 control).	Electric calendar software application for Android tablet.	Waiting list	Alzheimer’s disease (*n* = 12), cerebrovascular dementia (*n* = 5), senile dementia (*n* = 2), alcoholic dementia (n = 1), healthy older persons (*n* = 7). Mean age: 81.5 ± 6.9 years; mean MMSE 22 ± 4.0; 9 male, 18 female.	Semi-structured interviews with participants and their caregivers.	None	None
Park and Park 2018 (English). [33]	Korea. Community dwelling participants recruited via community welfare centers. No external funding received.	Single-blinded RCT. Measurements at baseline and after 10-week intervention; *n* = 78 (39 experimental; 39 control)	Non-specific computer training (NCT) with Nintendo Wii sports games.	Cognition-specific computer training (CCT) using the CoTras program.	MCI (MMSE NCT 26.41 ± 1.94, CCT group 26.67 ± 1.68); NCT group age 66.95 ± 4.10, CCT age 67.64 ± 4.55. 42 male, 36 female (NCT 20 male, 19 female; CCT 22 male, 17 female)	36-Item Short-Form Health Survey (SF-36) ^1^	SF-36 ^2^	
Pietilä et al. 2017 (Finnish). [34]	Finland. Community dwelling participants. Funding from the Miina Sillanpää Foundation.	Single-center, single-blinded, two-arm, RCT. Measurements at baseline, 13 weeks, and 6 months; *n* = 53 (28 experimental; 25 control)	FORAMEN rehab program, delivered by tablet computer.	Waiting list	Persons with early stage “mild” Alzheimer’s disease. Mean age 69.0 ± 5.0; 28 male, 27 female. Mean MMSE experimental group: 23.1 ± 3.8. Mean MMSE control group: 20.9 ± 3.8	Alzheimer’s Disease Cooperative Study Activities of Daily Living Scale (ADCS-ADL)	None	Beck Depression Inventory (BDI-II); World Health Organization Quality of Life BREF (WHOQOL-BREF); Care of Older People in Europe (COPE-index)
Silva et al. 2017 (English). [35]	United Kingdom. Community dwelling, recruited through day care centers. Funding from Region of Bourgogne (FABER); Fondation Médéric; and the Portuguese Foundation for Science and Technology.	Single-center, single-blinded, three-arm RCT. Measurements at baseline, after 6 weeks, and 6 months; *n* = 51 (17 experimental; 17 offline memory support; 17 control).	SenseCam wearable camera.	Memo+ arm: a pencil and paper cognitive training program. Control arm: daily activities diary	Diagnosed probable Alzheimer’s disease. Sensecam group mean age: 75.41 ± 5.26; mean MMSE 21.88 ± 3.33 Memo + group mean age: 71.71 ± 5.15; mean MMSE 21.53 ± 3.01 Control group: mean age 73.82 ± 5.74; mean MMSE 22.82 ± 1.85	Adults and Older Adults Functional Assessment Inventory (IAFAI).	World Health Organization Quality of Life OLD (WHOQOL-OLD) ^3^	None
Vanoh et al. 2019 (English). [36]	Malaysia. Community dwelling participants. Funding from the Ministry of Higher Education Malaysia.	Single-center, single-blinded, two-arm, RCT. Measurements at baseline and 6 months; *n* = 60 (30 experimental; 30 control).	WESIHAT 2.0© web-based wellness application.	Dietary counselling based on the “Healthy Eating” pamphlet produced by Ministry of Health Malaysia	Older adults with MMSE 16–28. Mean MMSE WESIHAT group 28.3 ± 1.78; mean MMSE control group 27.0 ± 2.63; mean age: 67.84 ± 5.65; 21 male, 29 female	World Health Organization Disability Assessment Scale (WHODAS 2.0) 12-item version ^4^	Medical Outcome Social Support Survey (MOSS); WHODAS 2.0 12-item version ^5^; three-item loneliness scale.	None

^1^ Role-emotional and role-physical subscales relevant. ^2^ Social functioning subscale relevant. ^3^ Social participation subscale relevant. ^4^ Mobility, self-care, and household/life activities subscales relevant. ^5^ Social/getting along and society/participation subscales relevant.

**Table 2 jcm-10-00604-t002:** Weight-of-evidence assessment of included studies.

Authors	WoE A *, Intrinsic Methodological Quality (Good, Fair, Poor)	WoE B, Appropriateness of Method in Context of this Review (High, Fair, Low)	WoE C, Relevance of Study to this Review Question (High, Fair, Low)	WoE D, Overall Assessment (High, Fair, Low)
Lee et al. [28]	Poor	Fair	Low	Low
Liao et al. [29]	Fair	Fair	Low	Fair
McCarron et al. [30]	Fair	Fair	Low	Fair
Mrakic-Sposta et al. [31]	Fair	Fair	Low	Fair
Nishihura et al. [32]	Poor	Low	Low	Low
Park and Park. [33]	Good	Low	Low	Fair
Pietilä et al. [34]	Fair	Fair	Low	Fair
Silva et al. [35]	Fair	High	Low	Fair
Vanoh et al. [36]	Fair	Fair	Low	Fair

* Assessed using the National Institute of Health (NIH) tool for the assessment of randomized controlled studies [26]. WoE: Weight-of-evidence.

**Table 3 jcm-10-00604-t003:** Results reported from included studies.

Authors	Outcome Measure	Results
Lee et al. [28]	HK-LIADL Scale (self-management)	Effect of intervention not statistically significant
Liao et al. [29]	LIADL (self-management)	Significant group and group by time interaction effects, in favor of the experimental group (group, *p* < 0.001, ƞ^2^p = 0.87; interaction, *p* < 0.01, ƞ^2^p = 0.217).
McCarron et al. [30]	PES-AD (social participation)	Effect of intervention not statistically significant
	DQoL (social participation)	Effect of intervention not statistically significant
Mrakic-Sposta et al. [31]	FAQ (self-management)	Effect of intervention not statistically significant
Nishihura et al. [32]	Semi-structured interviews (self-management)	SSI results not fully reported. Summary statement that positive self-management behavior changes were observed by caregivers in intervention group
Park and Park. [33]	SF-36 (self-management and social participation)	Role-emotional subscale: Significant group-by-time effect in favor of NCT (4.18 (95%CI 3.72 to 4.63), ƞ^2^p = 0.821 Role-physical subscale: Effect of intervention not statistically significant Social functioning subscale: Effect of intervention not statistically significant
Pietilä et al. [34]	ADCS-ADL (self-management)	Effect of intervention not statistically significant
	BDI-II (caregiver)	Effect of intervention not statistically significant
	WHOQOL-BREF (caregiver)	Effect of intervention not statistically significant
	COPE index (caregiver)	Subscales analyzed separately. Significant improvement in intervention group vs. control with respect to positive attitudes (*p* = 0.023). Effect size not reported. No statistically significant differences on other subscales.
Silva et al. [35]	IAFAI (self-management)	Global scores: statistically significant visit effect (F[2,43] = 16.26, *p* < 0.01, ƞ^2^p = 0.28) and group by visit interaction (F[2,43] = 8.71, *p* < 0.01, ƞ^2^p = 0.29). IADL familiar subscale: statistically significant visit effect (F[2,43] = 5.31, *p* < 0.01, ƞ^2^p = 0.11) and group by visit interaction (F[2,43] = 5.40, *p* < 0.01, ƞ^2^p = 0.21). IADL advanced subscale: statistically significant visit effect (F[2,43] = 11.74, *p* < 0.01, ƞ^2^p = 0.22) and group by visit interaction (F[2,43] = 4.83, *p* < 0.01, ƞ^2^p = 0.19). All reported significant effects demonstrated improvements in intervention group vs control
	WHOQOL-OLD * (social participation)	No subscale analyses for social participation reported. Overall, effect of intervention not statistically significant
Vanoh et al. [36]	WHODAS 2.0 (self-management and social participation)	No subscale analyses for social participation and self-management reported. Significant group, time, and interaction effects, in favor of the intervention group (group, *p* < 0.01, ƞ^2^p = 0.341; Time, *p* < 0.05, ƞ^2^p = 0.128; interaction *p* < 0.01, ƞ^2^p = 0.191)
	MOSS (social participation)	The four subscales were analyzed separately. Significant group by time interaction effects for informational support (*p* < 0.05, ƞ^2^p = 0.123) and tangible support (*p* < 0.01, ƞ^2^p = 0.186). No statistically significant group, time, or interaction effects with respect to positive social interaction or affective support
	Three-item loneliness scale (social participation)	Significant effect of group (*p* < 0.05, ƞ^2^p = 0.184) but no statistically significant time or interaction effects

* Social participation subscale relevant.

**Table 4 jcm-10-00604-t004:** Extracted statements from included studies regarding facilitators and barriers to implementation of the technological intervention.

Authors	Statements Concerning Facilitators in Implementing Intervention	Classification (Implementation, Mechanism of Impact, Context)	Statements Concerning Barriers in Implementing Intervention	Classification (Implementation, Mechanism of Impact, Context)
Lee et al. [28]	Qualitative feedback from participants showed that they enjoyed and liked the memory training program. They considered that learning to use a computer was not difficult. They would recommend the training program to others.	Mechanism of impact	More regular stimulating positive feedback could be integrated into the training program. It is suggested to increase the number of training sessions from 12 to 15.	Implementation
Liao et al. [29]	The enjoyment and attractiveness of VR characteristics may increase motivation and lead to extensive training effects, resulting in cognitive improvement.	Mechanism of impact	None identified	Not applicable (N.A.)
McCarron et al. [30]	None identified	N.A.	The majority of participants did not find the technology useful. Reasons: (1) complexity of the SSA, (2) enrollment process, (3) impracticality, (4) stigma, and (5) functionality of the SSA.	Mechanism of impact
Mrakic-Sposta et al. [31]	None identified	N.A.	None identified	N.A.
Nishihura et al. [32]	Eleven of the fourteen participants received sufficient support from their main caregiver. For instance, the caregiver visited them more than once a week to maintain the electric calendar.	Context	Some negative comments about the lack of durability of the battery and difficulties inputting the schedule. Three participants mentioned they did not need it in their daily lives.	Mechanism of impact
	Feedback interviews revealed that most participants had positive impressions of using the electric calendar.	Mechanism of impact		
Park and Park [33]	NCT can facilitate the training in a fun manner. Thus, NCT might motivate subjects more efficiently as compared to CCT, which may decrease issues with intervention compliance.	Mechanism of Impact	CCT had to be provided at a healthcare facility. Both training groups only received training individually, without opportunity for social interaction.	Implementation
Pietilä et al. [34]	None identified	N.A.	None identified	N.A.
Silva et al. [35]	SenseCam does not provoke negative evaluations, but suggests a beneficial action on mood. Authors refer to research suggesting that wearable cameras are less demanding than other external aids and are also less demanding and more motivating than paper and pencil cognitive exercises.	Mechanism of impact	None identified	N.A.
Vanoh et al. [36]	Group discussion was conducted to explain the proper ways to use the website, to describe the content in WESIHAT 2.0, and to identify the problems faced by subjects.	Implementation	Difficulties were reported by the intervention group in dealing with unknown people due to the fear of being deceived.	Implementation
	The counselling session was made interesting by having a quiz related to the content of WESIHAT 2.0 and playing free online brain games.	Mechanism of impact	The use of the compute requires complex motor functioning, language processing, and focus.	Mechanism of impact

## Data Availability

The data presented in this study are available in this article.

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
