# Peer review of "Can Use of Digital Technologies by People with Dementia Improve Self-Management and Social Participation? A Systematic Review of Effect Studies"

_jcm, 2021, doi:10.3390/jcm10040604_

Round 1

Reviewer 1 Report

The paper reports on scientific evidence for the use of digital technologies by people with dementia; a systematic review of controlled studies revealed only 8 studies that could be included. The authors chose to report the results by narrative review which seems appropriate with respect to the small number of studies included. The report includes all information required and its methodology is state of the art. The tables (especially table 4 and appendix) are rather lengthy and should be shortened.

Suggestions:

Introduction

  • include statements on mild cognitive impairment (MCI) in the introduction, because the studies integrated in this review also take into account subjects with MCI
  • line 81: one or two examples on the kind of interventions that are evaluated in this review would help the reader understand what to expect (the examples are only given in the results section)

Experimental section

  • line 210/211: "We did not plan to conduct any further statistical meta-analysis because of the expected heterogeneity (...)" - heterogeneity can be tested and controlled by meta-analytic methods, so this is no valid reason not to integrate the results by meta-analysis. A more valid reason for using a narrative review is the small number of studies included.

Results

  • table 1 does not give information on the treatment condition, i.e. which programs/devices were evaluated in the treatment group. Although this information is given in the text, I recommend adding it in table 1 due to its importance for the research question
  • table 3: study Nishihura et al, column 3: "SSI results not fully reported Summary statement (...)" - punctuation mark missing?
  • table 4 is rather lengthy: original statements concerning facilitators/barriers should be shortened or summed up by the authors

Appendix

  • the appendix needs formatting; in my opinion, the report of research strings is sufficient and the lengthy table can be deleted

Author Response

Reviewer 1

  1. Introduction: include statements on mild cognitive impairment (MCI) in the introduction, because the studies integrated in this review also take into account subjects with MCI

    Authors’ response: Thank you for your comment. We have included appropriate introductory statements regarding MCI as follows:
    Line 54-57: “Despite intensifying interest and investment, previous reviews of the literature have failed to consistently demonstrate high quality evidence for the effectiveness of technological solutions in improving cognitive functioning, self-management, or social participation for people with dementia or Mild Cognitive Impairment (MCI) [13-15].
    Lines 84-86: “A distinction can also be made between people experiencing impaired cognition with a diagnosis of dementia, and people with MCI. In this study, we are interested in both dementia and MCI.”
    Line 88, line 90, line 94 and line 102: “…dementia or MCI…”

    2. line 81: one or two examples on the kind of interventions that are evaluated in this review would help the reader understand what to expect (the examples are only given in the results section)

    Authors’ response: We agree that examples of what kind of interventions are evaluated will help readers understand the focus of the review better. We have added the following (lines 78-81): “Such technologies may include software applications or ‘apps’ designed to be used by the person with dementia, or include specific hardware components such as tablet computers, virtual reality devices, or wearable technologies.”

    3. Experimental section: line 210/211: "We did not plan to conduct any further statistical meta-analysis because of the expected heterogeneity (...)" - heterogeneity can be tested and controlled by meta-analytic methods, so this is no valid reason not to integrate the results by meta-analysis. A more valid reason for using a narrative review is the small number of studies included.

    Authors’ response: We agree that the small number of studies identified is also an important reason why narrative synthesis was more appropriate than statistical meta-analysis in this review. We agree that diversity of outcome measures, for example, can be dealt with within the scope of statistical meta-analysis. However, the heterogeneity with respect to the nature and primary aim of the intervention was also a factor in our decision to proceed with narrative synthesis. We have edited the manuscript accordingly. Line 223-226: We removed the phrase, “…because of the expected heterogeneity of the interventions and outcome measures included in this review, which we feel would render meta-analysis clinically meaningless.” Line 223-224: We added the phrase, “…because of the expected heterogeneity in technological interventions, and small number of included studies.”

    4. Results: table 1 does not give information on the treatment condition, i.e. which programs/devices were evaluated in the treatment group. Although this information is given in the text, I recommend adding it in table 1 due to its importance for the research question

    Authors’ response: Thank you for your comment. We have added a column to Table 1 naming the experimental group intervention. In order to avoid making the table too long and to avoid duplicating the description in the main body of the manuscript, we have kept this brief. See Table 1, page 7-10

    5. table 3: study Nishihura et al, column 3: "SSI results not fully reported Summary statement (...)" - punctuation mark missing?

    Authors’ response: thank you, we have added the correct punctuation. See Table 3, page 13-14. 6. table 4 is rather lengthy: original statements concerning facilitators/barriers should be shortened or summed up by the authors Authors’ response: we have reviewed the quoted statements and where possible shortened and summarized these, such that the length of the table has been reduced to almost half of the original length. See Table 4, page 15-17.

    7. Appendix: the appendix needs formatting; in my opinion, the report of research strings is sufficient and the lengthy table can be deleted.

    Authors’ response: thank you for the suggestion. As suggested, we have deleted the tables as suggested (page 22-25, page 27-96, page 98-169, page 170-174) and left the strings only.

Reviewer 2 Report

This systematic review investigated the therapeutic effects of digital technologies used by people with dementia, to improve self-management and social participation. Eight studies were included from four digital databases. The authors claimed that there is no high-quality evidence about the effectiveness of the three technologies (virtual reality, wearables and software applications) for improving self-management or social participation when used by people with dementia. Thus, practitioners and caregivers should decide on the usage of digital technologies for the management of dementia case-by-case.

Generally speaking, it is an interesting topic and the manuscript was well written. However, some concerns existed.

  1. According to the journal’s guideline, a systematic review should be reported adhering to the Preferred Reporting Items for Systematic Review and Meta-analysis (PRISMA) checklist. Thus, a statement about following the PRISMA checklist should be added before Section 2.1 and the checklist could be found in supplementary files. Also, what handbook was used for this systematic review should be added before Section 2.1.
  2. The following judgments were based on the PRISMA checklist items.
  • Methods – item 5: The authors stated that they did not publish a protocol for this review, however, why the authors did not register their protocol in a registry such as PROSPERO. It is important to register their protocol before a systematic review has been done, since ‘Without review protocols, how can we be assured that decisions made during the research process aren’t arbitrary, or that the decision to include/exclude studies/data in a review aren’t made in light of knowledge about individual study findings? (http://www.prisma-statement.org/Protocols/WhyProtocols)’. The authors should explain why they did not register their protocol in the discussion section as well.
  • Methods – item 6: As the authors reported that they included articles published in any language, the language of the included studies should be reported in the characteristics table. Additionally, the authors did not mention what they did if an included study was published in a language other than English or their native languages. Did the authors recruit a qualified translator to translate the papers into English?
  • Methods – item 7: Databases with dates of coverage should be mentioned in the method section.
  • Methods – item 11: Funding sources of the included studies should be reported.
  1. It is important to note that only four databases were searched. Although the authors claimed that it is a limitation of this study, it does not make sense to me why they did not search more databases, such as Science Direct, Scopus, Cochrane Central Register of Controlled Trials, ProQuest, AMED, and Google Scholar. It is not an unsolvable challenge. Also, the authors mentioned that one of the strengths of this study was to perform a systematic search. Could you really claim that it was a systematic search in the only four databases, not even included databases in other languages, such as China Academic Journals (CNKI), IndMed, Latin American and Caribbean Health Sciences (LILACS). Therefore, I suggested that more databases should be included.
  2. References are needed for the excluded studies (lines 231 to 239).
  3. Section 2 should be ‘Materials and method’. 
  4. Suggest putting Appendix 1 to supplementary files.

Author Response

Reviewer 2

Generally speaking, it is an interesting topic and the manuscript was well written. However, some concerns existed.

1. According to the journal’s guideline, a systematic review should be reported adhering to the Preferred Reporting Items for Systematic Review and Meta-analysis (PRISMA) checklist. Thus, a statement about following the PRISMA checklist should be added before Section 2.1 and the checklist could be found in supplementary files. Also, what handbook was used for this systematic review should be added before Section 2.1.

Authors’ response: Thank you for your comment. The PRISMA checklist has already been submitted as a supplementary file. We have added a statement to the effect that we used the PRISMA checklist and cited the best-practice guidance as followed (line 106-107): “This systematic review was carried out with reference to the PRISMA guidance, and best practice as outlined by the Centre for Reviews and Dissemination [23].”

2. Methods – item 5: The authors stated that they did not publish a protocol for this review, however, why the authors did not register their protocol in a registry such as PROSPERO. It is important to register their protocol before a systematic review has been done, since ‘Without review protocols, how can we be assured that decisions made during the research process aren’t arbitrary, or that the decision to include/exclude studies/data in a review aren’t made in light of knowledge about individual study findings? (http://www.prisma-statement.org/Protocols/WhyProtocols)’. The authors should explain why they did not register their protocol in the discussion section as well.

Authors’ response: The authors understand the important advantages of registering systematic review protocols. We prepared the protocol for this review in April 2020. At that time, we wished to register the protocol with PROSPERO. Due to COVID-19 disruption, PROSPERO was experiencing delays of unknown duration in registering protocols but was still advising that protocols should only be registered before the review began. Given the relevance and urgency of this review in light of COVID-19 and we did not want to risk significantly delaying the work by postponing data extraction for an indefinite period of time whilst waiting for registration. That is why we chose to proceed with the review, without registering the protocol. We did not take the decision lightly, and we assure the reviewers that we have held true to the protocol developed. However, we felt that we had to take the pragmatic approach, and trust in the quality of the peer review process to assess the work as presented, in good faith.

3. Methods – item 6: As the authors reported that they included articles published in any language, the language of the included studies should be reported in the characteristics table. Additionally, the authors did not mention what they did if an included study was published in a language other than English or their native languages. Did the authors recruit a qualified translator to translate the papers into English?

Authors’ response: Thank you for your comment. We have added the language in the first column of Table 1 and added a clarifying statement to the methods section, as follows (lines 175-176): “Where required, translation from languages other than English and Dutch was carried out by qualified translators.”

4. Methods – item 7: Databases with dates of coverage should be mentioned in the method section.

Authors’ response: Thank you for this comment. We have updated the methods section accordingly (lines 109-112): “Searches were conducted of the following databases, within the date range 01/01/2007 to 09/04/2020: Pubmed, APA PsycInfo, Web of Science, CINAHL and the Cochrane Central Register of Controlled Trials.”

5. Methods – item 11: Funding sources of the included studies should be reported.

Authors’ response: Thank you for your comment. We have modified Table 1 to include this information in the second column (page 7-10).

6. It is important to note that only four databases were searched. Although the authors claimed that it is a limitation of this study, it does not make sense to me why they did not search more databases, such as Science Direct, Scopus, Cochrane Central Register of Controlled Trials, ProQuest, AMED, and Google Scholar. It is not an unsolvable challenge. Also, the authors mentioned that one of the strengths of this study was to perform a systematic search. Could you really claim that it was a systematic search in the only four databases, not even included databases in other languages, such as China Academic Journals (CNKI), IndMed, Latin American and Caribbean Health Sciences (LILACS). Therefore, I suggested that more databases should be included.

Author response: Thank you for your comment. The databases that we chose to search – PubMed, CINAHL, APA Psychinfo and Web of Science – were carefully chosen for: 1) their relevance to the search question, 2) their likely value to answering this review question and 3) feasibility, with respect to the time and resources available for the researchers.
We were specifically interested in identifying controlled studies and we therefore agree with the reviewer that it is relevant, valuable and feasible to extend the search to the Cochrane Central Register of Controlled Trials, which we have now done. In the context of screening the trial registry results, we also found a relevant, valuable and feasible application of Google Scholar. With respect to other databases suggested: we do not think that it would provide significant additional value to conduct searches using Science Direct (which is not a database of academic journals, but rather a publishers’ platform), ProQuest (which is a search engine rather than a database per se), or Scopus (which is a very similar database to Web of Science, of which we already conducted a search). With regard to databases in other languages, this is the issue which remains a limitation of the study, as we originally noted. Whilst we have not excluded results that were listed in the databases searched, based solely on the language in which they were published, it is not feasible for our team, with the resources available to us to extend the scope of this review to conduct an exhaustive search of all foreign language databases. We have revised the discussion of the strengths and limitations of the review to reflect our approach to non-English language publications, to clarify our position (lines 560-564):
“A limitation of this review is that the execution of the search strategy was limited to English-language databases, although we did not exclude any of the articles we identified just because they were not published in English. We did not include grey literature in this review, nor or other possible sources of results. We did not undertake a specific effort to identify possible unpublished results.”
Through the additional search with the Cochrane Central Register of Controlled Trials and Google Scholar, we identified an additional 104 unique records, only one of which was selected for full-text screening. This result met the inclusion criteria for this review (reference 33). The method, results and discussion sections of the article, and the associated figures have been amended accordingly. Our conclusions have not changed based on the additional result.

Lines 109-112: “A systematic search strategy was developed by DN and CP. Searches were conducted of the following databases, within the date range 01/01/2007 to 09/04/2020: Pubmed, APA PsycInfo, Web of Science, CINAHL and the Cochrane Central Register of Controlled Trials.”

Lines 117-120: “Where trial registry entries were identified but there were no published results listed in the registry, a further search using Google Scholar was conducted, using the trial registration number as the search term, and any articles identified were screened for inclusion in the review.”

Throughout the results section (line 234-434), the numbers of results identified, screened, excluded at each step and the final results of included results have been changed.

Figure 1 has been amended with the new total numbers of results (page 6).

Tables 1-4 have been updated to include the results of the new included study, reference 33 (pages 7-10, 12, 13-14 and 15-17).

Lines 290-294: “VR-based non-specific computer training [33]: The aims of the intervention were to improve users’ cognitive function and quality of life. Users were trained to use the Nintendo Wii to engage in virtual reality sports activities, including table tennis, fencing and archery. The intervention was delivered by occupational therapists, during three sessions per week for ten weeks, of 10 minutes duration.”

Throughout the discussion section (line 435-621), references to the number of results included, clustered and discussed with respect to various characteristics have been changed.

7. References are needed for the excluded studies (lines 231 to 239).

Authors’ response: Thank you for your comment. We have added the appropriate references, numbered 37 to 74.

8. Section 2 should be ‘Materials and method’.

Authors’ response: thank you for your comment. In preparing this manuscript we used the template provided by JCM and did not change the headings. In line with the suggestion, we have amended the section 2 heading to read ‘Materials and method’ (line 105), pending confirmation from the Editor that this is acceptable.

9. Suggest putting Appendix 1 to supplementary files.
Authors’ response: thank you for the suggestion. Please see our response to Reviewer 1. We have removed the tables and reported the search strings only. We are very happy to further move the search strings to “Supplementary files”.

Round 2

Reviewer 2 Report

The authors modified the manuscript according to the reviewer's instructions.

However, I still believe that it is better to move the extensive Appendix A to supplementary files.

Author Response

Dear Marina Tukovic,

Thank you for sharing the feedback of the reviewers on the revision of this manuscript. Below, I have responded to the comment of the reviewer, and I have made the appropriate changes to the submission.

I hope that on the basis of this minor revision that the article can be accepted for publication in JCM.

With kind regards,

David Neal (on behalf of the authors)

Response to reviewers

Reviewer 2:

"I still believe that it is better to move the extensive Appendix A to supplementary files."

Authors' response:

Thank you for your comment. We have removed the appendi from the manuscript and submitted this as a seperate file under supplementary materials.